# Adversarial Inputs for Linear Algebra Backends

**Jonas Möller** [1,2]  **Lukas Pirch** [1,2]  **Felix Weissberg** [1,2]  **Sebastian Baunsgaard** [1,2]
**Thorsten Eisenhofer** [1,2]  **Konrad Rieck** [1,2]

## Abstract

Linear algebra is a cornerstone of neural network inference. The efficiency of popular frameworks, such as TensorFlow and PyTorch, critically depends on backend libraries providing highly optimized matrix multiplications and convolutions. A diverse range of these backends exists across platforms, including Intel MKL, Nvidia CUDA, and Apple Accelerate. Although these backends provide equivalent functionality, subtle variations in their implementations can lead to seemingly negligible differences during inference. In this paper, we investigate these minor discrepancies and demonstrate how they can be selectively amplified by adversaries. Specifically, we introduce *Chimera examples*, inputs to models that elicit conflicting predictions depending on the employed backend library. These inputs can even be constructed with integer values, creating a vulnerability exploitable from real-world input domains. We analyze the prevalence and extent of the underlying attack surface and propose corresponding defenses to mitigate this threat.

## 1. Introduction

Frameworks like TensorFlow and PyTorch provide a high-level interface to machine learning, enabling developers to deploy models across diverse platforms. These frameworks abstract away the complexities of low-level implementations and hardware, offering unified access to the computing resources of each platform, ranging from large clusters to mobile devices and embedded systems. A cornerstone of this abstraction lies in linear algebra backends, which deliver optimized vector and matrix operations tailored to the peculiarities of each platform, such as dot products, rank updates, matrix multiplications, and convolutions.

[1]Berlin Institute for the Foundations of Learning and Data (BIFOLD), Germany [2]TU Berlin, Germany. Correspondence to: Jonas Möller <jonas.moeller.1@tu-berlin.de>.

*Proceedings of the $42^{nd}$ International Conference on Machine Learning*, Vancouver, Canada. PMLR 267, 2025. Copyright 2025 by the author(s).

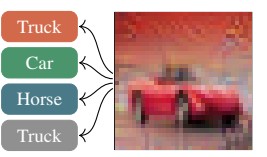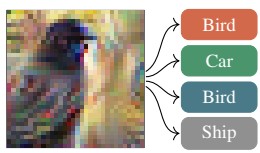

*Figure 1.* Chimera examples: Conflicting predictions made by the same model, depending on the used linear algebra backend such as Apple Accelerate (⬤), Intel MKL (⬤), Nvidia CUDA (⬤), BLIS (⬤).

Technically, this abstraction builds upon the classic BLAS specification by Lawson et al. (1979), which introduces a standardized interface for linear algebra, referred to as *Basic Linear Algebra Subprograms.* While linear algebra backends strictly adhere to this specification, their low-level implementations vary significantly based on the targeted platform, encompassing the choice of algorithms, parallelization strategies, memory management, and hardware support (Dongarra et al., 1990). Consequently, a wide array of these backends has emerged in practice, including Intel MKL, Nvidia CUDA, Apple Accelerate.

Due to these differences, no linear algebra library backend behaves exactly like another, and subtle deviations regularly occur when processing the same input. These discrepancies arise from the inherent fragility of floating-point arithmetic: as float representations only approximate real numbers, operations such as addition and multiplication are not strictly associative (Goldberg, 1991). For example, when multiplying two $1000 \times 1000$ matrices sampled uniformly from $[0, 1)$ with 32-bit precision, deviations of approximately $6 \cdot 10^{-10}$ can be observed between backends (see Section 2). Hence, the inference of learning models inevitably leads to minor inaccuracies in their predictions.

Given the minuscule scale of these differences, it is tempting to assume that backends provide sufficiently similar computations for most practical applications. In this paper, we challenge this assumption and pose the question: Is it possible to craft an input for a model that elicits conflicting predictions depending on the employed backend library? We refer to these inputs as *Chimera examples*, as they exhibit differing appearances depending on the backend library used. Figure 1 illustrates examples of these inputs, each resulting in three different predicted classes depending on the linear algebra backend employed.

So far, previous work has focused on floating-point imprecision arising from differences in CPU architectures, for example, for fingerprinting systems (Schlögl et al., 2021; 2024) or breaking the certification of models (Jin et al., 2022; 2024; Voráček & Hein, 2023). Our analysis of linear algebra backends builds on this work; yet, we aim to induce significantly larger changes that flip the prediction of a model given an adversarial input. While differences in CPU architecture may further exacerbate this issue, we demonstrate that Chimera examples also exist between backends on the same CPU architecture.

Uncovering Chimera examples for a model, however, is surprisingly challenging: First, the exploited minor differences between backends are non-continuous and non-differentiable, obstructing the use of common attack strategies. Second, the input to models cannot be assumed to have arbitrary precision. For example in computer vision, images are typically represented by 8-bit or 16-bit matrices, making it hard to control tiny differences in calculations. To address these challenges, we present a generalized method for constructing Chimera examples. Using this method, we explore the underlying attack surface and evaluate its prevalence across six backend libraries, including all major platforms. Finally, we propose a defense that mitigates the threat of Chimera examples.

In summary, we make the following major contributions:

- We present the first method for constructing adversarial inputs that induce conflicting predictions of models due to differences in linear algebra backends.

- We identify and analyze Chimera examples across six common backends, including GPU-accelerated and CPU-based implementations.

- We derive a defense mechanism for Chimera examples based on our analysis, preserving both the model's accuracy and its deterministic nature.

## 2. Background

Subtle inconsistencies observed in the output of linear algebra backends stem from the inherent limitation of representing real numbers, $\mathbb{R}$, within a fixed bit representation. To set the scene for our analysis, we first examine how floating-point numbers approximate $\mathbb{R}$ and then explore how variations in the implementation of linear algebra operations can magnify these deviations during computation.

### 2.1. IEEE-754 Floats

The IEEE-754 specification (IEEE, 1985; 2019) serves as the de-facto standard for floating-point numbers $\mathbb{F}$, commonly referred to as floats. Widely adopted, the standard

specifies binary formats for 16-bit, 32-bit, 64-bit, and 128-bit representations. In each format, floats are represented as triples consisting of a sign $s$, an exponent $e$, and a significand $m$, where the exponent uses $w$ bits and the significand uses $t$ bits. A real number $x \in \mathbb{R}$ can then be approximated using $b = 2^{(w-1)} - 1$ and $p = t - 1$ by:

$$x \approx (-1)^s \cdot 2^{(e-b)} \cdot (1 + 2^{(1-p)} \cdot m).$$

Due to their fixed bit representation, floats can naturally represent only a finite subset of $\mathbb{R}$. As a consequence, common properties of arithmetic in $\mathbb{R}$ do not hold for floats. For example, IEEE (1985) specifies that $\mathbb{R}$ values are correctly rounded if they are mapped to the nearest representable value in $\mathbb{F}$. This rounding scheme introduces non-associativity in operations, such as $(a + b) + c \neq a + (b + c)$, since all intermediate results are rounded to the nearest representable value. While the BLAS specification defines the inputs and outputs of matrix operations, it does not specify the order of computational steps, creating a vulnerable spot via this non-associativity.

To analyze the differences resulting from this imprecision, we quantify the distance between floats using the concept of *units in the last place* (*ULP*) (Goldberg, 1991). In the discrete space of $\mathbb{F}$, this metric measures the number of representable values between two floats, independent of the exponent and scale of the numbers. For instance, the 32-bit floats representing the numbers 1 and 1.0000001, as well as 10,000,000 and 10,000,001, both have a distance of only 1 ULP. Practically, for same-sign floats the measure is easy to calculate by casting the bit representations of the floats to integers and returning the absolute delta.

### 2.2. BLAS Interface

Similarly to IEEE-754, BLAS is the de-facto standard for linear algebra operations in machine learning. Lawson et al. (1979) introduced the term BLAS, referring to 38 subprograms for linear algebra operations such as dot products, matrix multiplications, and convolutions. Since its introduction, the specified set of operations has undergone numerous updates and extensions (Blackford et al., 2002) and is now implemented in various backends, including Intel MKL, Nvidia CUDA, and Apple Accelerate.

To allow a high degree of freedom in implementation, the BLAS specification states: *"any algorithm that produces results close enough to the usual algorithms presented in a standard book on matrix computations is acceptable"* (Blackford et al., 2002). This flexibility is crucial for accommodating advanced computation techniques, such as the Strassen (1969) algorithm. However, this same flexibility in implementation leads to variations in results across different BLAS backends.

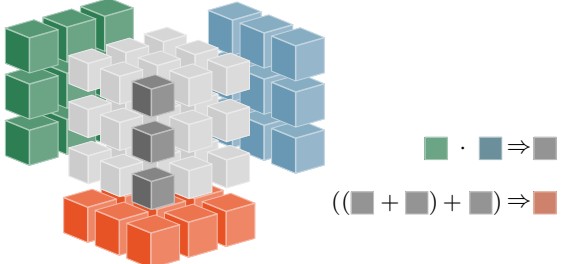

*Figure 2.* Abstract visualization of GEMM, $\mathbf{C} = \mathbf{AB}$. The sum of a box column in the middle cube symbolizes the addition into a single block of $\mathbf{C}$. Because arithmetic in $\mathbb{F}$ is non-associative, the order of adding column boxes matter.

## 2.3. Sources of Differences

To illustrate the impact of this flexibility, we consider matrix multiplication as an example. Specifically, we have three matrices $\mathbf{A}$, $\mathbf{B}$, and $\mathbf{C}$, with a general matrix multiplication (GEMM) operation defined as $\mathbf{C} = \mathbf{AB}$. The most common approach to implementing GEMM efficiently involves decomposing $\mathbf{A}$ and $\mathbf{B}$ into smaller blocks and distributing the computation, as also recommended in the BLAS specification (Blackford et al., 2002).

In particular, the matrices $\mathbf{A}$ and $\mathbf{B}$ are divided into small blocks $\mathbf{X} \in \mathbb{F}^{\varrho \times \varrho}$, whose size $\varrho$ is determined by the available cache of the underlying hardware (Goto & van de Geijn, 2008a). Depending on CPU features, such as SSE and AVX, these blocks are multiplied using hardware-accelerated kernels that concurrently process multiple elements within each block. As an example, a 32-bit float GEMM multiplication in OpenBLAS uses a 16x4 kernel[1] and 1024x1024 blocks on an Intel Xeon Gold processor.

Figure 2 illustrates this decomposition process. The matrices $\mathbf{A}$ and $\mathbf{B}$ are divided into blocks, whose products are sequentially accumulated in $\mathbf{C}$. While each block multiplication is accurate within the limits of IEEE 754, the accumulation in $\mathbf{C}$ amplifies deviations due to non-associativity. Depending on the kernel size, block size and processing order, the backends implicitly prioritize the summation of values into $\mathbf{C}$, which can be conceptually visualized by adding parentheses to group additions.

Note that GEMM is just one example, as many other operations similarly suffer from rounding errors. Two prominent cases include aggregation (Kahan, 1965), commonly used in machine learning for gradient summation, and convolutions, which can be interpreted or transformed into GEMM calls (Chellapilla et al., 2006). Like GEMM, efficient implementations of both operations compute results using blocks and kernels.

---

[1]OpenBLAS 16x4 32-bit float kernel.

## 3. Attacking BLAS Backends

Equipped with an understanding of deviation sources, we are now ready to tackle the generation of adversarial inputs exploiting these inconsistencies. In particular, we focus on inputs that cause a model to produce conflicting predictions. Before outlining our approach, we first introduce notation and formalize the concept of Chimera examples.

### 3.1. Chimera Examples

We consider a learning model $\theta$ for classification, deployed across $n$ different platforms, each utilizing a distinct linear algebra backend. The inference process of $\theta$ on these backends gives rise to $n$ functions, $f_1, \ldots, f_n : \mathbb{F}^d \to \mathbb{F}^c$, where $d$ is the number of input dimensions and $c$ the number of output classes. Due to variations in backend implementations, each pair of functions $f_i, f_j$ may exhibit slight deviations in its output, creating an attack surface.

In practice, models are rarely applied over the entire range of floating-point numbers $\mathbb{F}^d$. Instead, the input format constrains the data to a discrete set of feasible vectors, which we denote as $\mathbb{S} \subset \mathbb{F}^d$. For example, when processing an image, individual pixels are represented as fixed-bit integers, drastically reducing the input domain. Similarly, when processing text, inputs are inherently discrete and map only to a small subset of the space represented by $\mathbb{F}^d$.

As a consequence, we consider the function $f_i$ as part of a high-level classification function

$$h_i : \mathbb{S} \to \{1, \ldots, c\}, \quad x \mapsto \arg\max_k f_i(x)_k,$$

which receives an input from $\mathbb{S}$ and predicts a class label. Based on this function, we can formally define Chimera examples that exist in the input domain $\mathbb{S}$.

**Definition 3.1** (Chimera example). Let $h_1, \ldots, h_n$ be a set of classification functions that utilize the same model $\theta$ on $n$ different linear algebra backends. A *Chimera example* is an input $\bar{x}$ that satisfies

$$h_i(\bar{x}) \neq h_j(\bar{x}), \quad \forall i \neq j, \quad \bar{x} \in \mathbb{S},$$

so that the predictions conflict for every pair $i, j$ of the considered backends.

The discrete nature of $\mathbb{S}$ plays a crucial role in the existence of Chimera examples and has received little attention in current literature on numerical imprecision. While differences on the scale of a few ULPs can be easily induced using high-precision floats, achieving similarly fine-grained variations from within a limited set $\mathbb{S}$ is significantly more challenging. Consequently, the attack surface depends not only on the deviations between backends but also on those that can be realized in practice. This challenge is illustrated in Figure 3, which shows deviations between the functions $f_1$ and $f_2$ while overlaying the feasible vectors of $\mathbb{S}$.

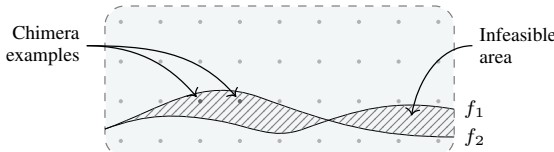

*Figure 3.* Schematic depiction of Chimera examples. The light blue area represents $\mathbb{F}^d$ ($\bullet$), while feasible vectors in $\mathbb{S}$ are shown as dots ($\bullet$). Chimera examples are feasible vectors that lie within a pocket between the decision boundaries of $f_1$ and $f_2$.

### 3.2. Threat Model

We assume an adversary with white-box access to the learning model $\theta$ and the employed linear algebra backends. This access enables them to perform inference using the classification functions $h_i, \ldots, h_n$, analyze intermediate computations such as $f_i(x)$, and compute gradients over $\theta$ on individual backends. Note, however, the adversary does not have direct access to genuine numerical results themselves, as also their computations must be executed on some linear algebra backend.

This threat model reflects common scenarios where models are deployed in different configurations in practice, known to the adversary. These configurations may include development and production systems, as well as high-performance and mobile/embedded implementations. While white-box access is not typically guaranteed in such settings, we adopt this strong attacker model to better evaluate the corresponding defenses introduced later (Section 5).

One may argue that keeping the models and backends confidential might be a possible defense, since black-box attacks are significantly harder to conduct in this brittle environment. However, this is not a reliable protection strategy, as it depends on keeping information confidential that is typically not considered sensitive. Moreover, in standard machine learning frameworks, the available backends are usually known to the adversary by default.

### 3.3. Finding Chimeras

For constructing inputs following Definition 3.1, we build on a common strategy for generating adversarial examples (Carlini & Wagner, 2017): We first select a starting point $x_1$ and iteratively move it toward the decision function by computing a perturbations from its gradient,

$$\delta = \nabla_{x_k} \ell\left(f_i(x_k), y_i\right),$$
$$x_{k+1} = x_k + \alpha\delta,$$

where $\ell$ is a loss function, such as the cross-entropy loss, $\alpha \in \mathbb{R}^+$ the step size, and $y_i$ the target label for the linear algebra backend $i$.

For a single backend $i$ this formulation moves the sample from its source class to a target class $y_i$. If we dynamically adapt the step size $\alpha$, this approach theoretically brings us infinitesimally close to the decision boundary, potentially leading to deviations among multiple backends.

However, there is a catch: the generated points do not lie within $\mathbb{S}$, and thus moving along their gradients may lead into infeasible regions, as demonstrated in Section 4.5. To address this problem, we introduce two strategies: First, we map $x_k$ back to $\mathbb{S}$ when computing its gradient, ensuring that the gradients reflect the view from $\mathbb{S}$ while optimization occurs in $\mathbb{F}$. Second, we let the backends "compete" against each other. This represents an important distinction from the standard search for adversarial examples. By setting a different target label $y_i$ for each backend $i$, we achieve a tug-of-war effect. Because each backend pulls the input towards its target class, the input sample is ultimately moved closer to the decision boundary.

Considering these strategies and multiple backends, we arrive at the following aggregated perturbation:

$$\delta = \frac{1}{n}\sum_{i=1}^{n}\nabla_{\bar{x}_k}\ell\left(f_i(\bar{x}_k), y_i\right)$$
$$\text{with } \bar{x}_k = q(x_k) \text{ and } \forall_{i \neq j} \, y_i \neq y_j$$

where $n$ is the number of backends and $q : \mathbb{F}^d \to \mathbb{S}$ maps an input from the floating point space $\mathbb{F}^d$ back to the constrained input space $\mathbb{S}$. Depending on the input domain, this function can be implemented as a quantization, an assignment to a grid, or a lookup of the closest elements in $\mathbb{S}$. In the image domain, the quantization function $q$ maps the input vector to the closest vector representing a valid 8-bit image. One step of this search method, including quantization, is illustrated in Figure 4.

To ensure that the input $x_k$ remains within the general bounds of the input domain, i.e., for images in $[0, 1]$, we also introduce a change of variables during optimization, similar to Carlini & Wagner (2017), reparameterizing each instance as $x_k = \frac{1}{2}(\tanh(w_k) + 1)$. For brevity, we omit the additional variable $w$ from the mathematical notation.

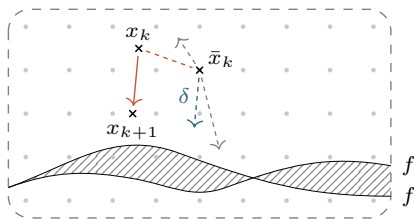

*Figure 4.* Perturbation step of our method. The point $x_k$ is moved along the gradient determined from a feasible input $\bar{x}_k$.

The resulting method is described in Algorithm 1. We search for an input $\bar{x}_k \in \mathbb{S}$ that satisfies the Chimera conditions (Definition 3.1). The loop terminates when a Chimera is found or the maximum iterations $N = 3000$ is reached. Note that we express the calculation of the aggregated perturbation as a for-loop, as it depends on an architecture capable of simultaneously obtaining results from multiple backend instances, such as virtual machines or containers. The starting point $x_1$ is obtained from 2000 iterations of our search on a single backend to move towards the proximity of the decision boundary first.

---

**Algorithm 1** Finding Chimera examples

**Input:** Starting point $x_1$; $f_i$ and $y_i$ for each backend $i$
**Result:** Chimera example $\bar{x} \in \mathbb{S}$

**for** $k \leftarrow 1$ **to** $N$ **do**
    $\bar{x}_k = q(x_k)$
    **if** $\bar{x}_k$ *is a Chimera* **then**
        $\lfloor$ **break**
    $\delta = 0$
    **for** $i \leftarrow 1$ **to** $n$ **do**
        $\lfloor$ $\delta \leftarrow \delta + \nabla_{\bar{x}_k} \ell(f_i(\bar{x}_k), y_i)$
    $x_{k+1} = x_k + \frac{\alpha}{n}\delta$
**return** $\bar{x}_k$

---

# 4. Chimeras in Practice

We begin our empirical evaluation by investigating the existence of Chimera examples in practice. Our goal is to assess whether we can construct corresponding feasible inputs for a learning model and sufficiently amplify their effect to cause conflicting predictions.

To facilitate future work, we have uploaded our source code to https://github.com/mlsec-group/dila

## 4.1. Linear Algebra Backends

For our experiments, we consider the linear algebra backends shown in Table 1, which are widely used in machine learning frameworks, such as TensorFlow and PyTorch.

*OpenBLAS* (Xianyi et al., 2012) is an open-source, CPU-focused library and the successor to the discontinued GotoBLAS implementation (Gunnels et al., 2001; Goto & van de Geijn, 2008b). It is a prominent BLAS backend with numerous CPU-specific optimizations and is widely used in projects such as NumPy, SciPy, and OpenCV.

*BLIS* (Van Zee & van de Geijn, 2015) is another open-source backend for linear algebra, developed in an academic setting. It provides a superset of BLAS functionalities while maintaining a high-performance base interface.

*Eigen* is free software and a widely used C++ library that implements the BLAS interface, providing efficient linear

*Table 1.* Overview of considered linear algebra backends

| Backend | Type | Platform | Version |
|---|---|---|---|
| OpenBLAS | CPU | Cross-platform | 0.3.28+ |
| Eigen | CPU | Cross-platform | 1.0 |
| BLIS | CPU | Cross-platform | 3.4.0+ |
| Intel MKL | CPU | Cross-platform | 2024.1.0-691 |
| Apple Accelerate | CPU | Apple Silicon | macOS 14.6.1 |
| Nvidia cuBLAS | GPU | Nvidia GPUs | 12.4 |

algebra operations. Eigen supports a wide range of CPU features and extensions.

*Intel MKL* (Intel, 2025) (Math Kernel Library) is a proprietary library developed by Intel, offering an extensive set of highly optimized kernels. Despite its origin, the backend is not restricted to Intel CPUs and provides cross-platform functionality with a BLAS interface.

*Nvidia cuBLAS* is a BLAS implementation included in the installation of CUDA (NVIDIA, 2020), a library for GPU calculations. It uses massively parallel instructions to compute linear algebra. We use CUDA 12.4.

*Apple Accelerate* (Apple, 2025) contains the default collection of operations on macOS, featuring a BLAS implementation. It utilizes either the on-die CPU or the GPU via Metal Performance Shaders (MPS). We found that the GPU-based version of MPS does not behave deterministically and so we disabled this extension of the library.

## 4.2. Experimental Setup

We consider three datasets, FMNIST (Xiao et al., 2017), CIFAR-10 (Krizhevsky et al., 2009), and ImageNet (Deng et al., 2009). For FMNIST, we use a fully connected network with two layers. For CIFAR, we employ a convolutional neural network with three VGG blocks (Simonyan & Zisserman, 2015) and three dense layers. We train both models to achieve a test accuracy of $82.32\,\%$ and $80.75\,\%$, respectively. For ImageNet, we use a pre-trained EfficientNetV2S (Tan & Le, 2021) with a test accuracy of $84.2\,\%$. Refer to Appendix B for more details. We use 32-bit floats with 23-bit significand in all experiments.

For all of our experiments, we execute the same code on top of PyTorch v2.5.1 with the different BLAS backends. All libraries use the default number of threads, as would be employed in a practical scenario. Moreover, we conduct our experiments on the following three platforms:

**P1** an Intel Xeon Gold 6326 CPU @ 2.90GHz, 16 cores, and 24 MB L3 cache (Ice Lake),

**P2** an Intel Xeon Silver 4114 CPU @ 2.20GHz with an Nvidia RTX 3090 24GB GPU

**P3** a Macbook Air M2, running macOS Sonoma 14.6.1

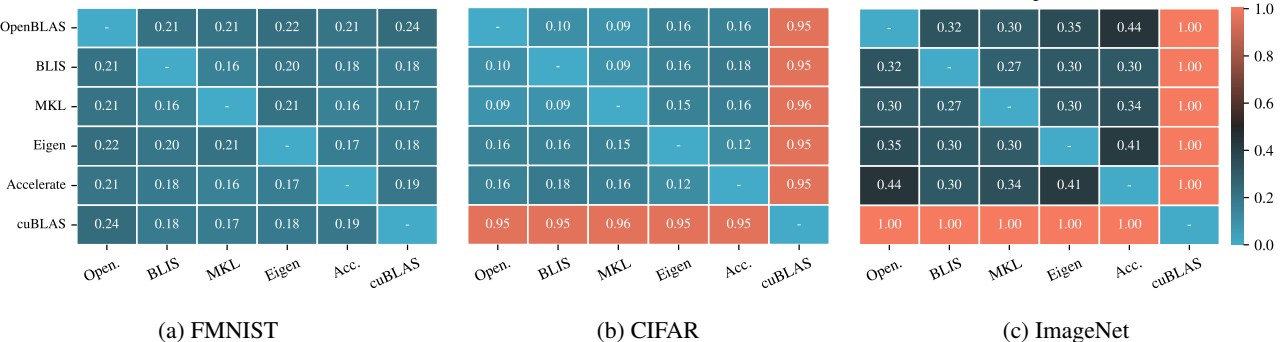

*Figure 5.* Attack success rates for each pair of BLAS libraries on FMNIST, CIFAR and ImageNet.

Except when the Apple Accelerate or Nvidia cuBLAS backends are considered, all experiments use platform P1. Otherwise, the respective platforms P2 and P3 are used. This experimental setup minimizes the impact of CPU differences, as all backends run on the same CPU architecture, except for Apple Accelerate and Nvidia cuBLAS, which are bound to specific hardware.

Due to blocking and other design choices in efficient matrix multiplication (see Section 2), the size and order of mini-batches introduce extra sources of indeterminism, even within equivalent BLAS implementations. To mitigate this effect, and to ensure the reproducibility of our experiments, we default to inference batches of size one.

### 4.3. Finding Chimeras

To uncover Chimera examples, we systematically search across all six linear algebra backends. Specifically, we consider pairs of backends ($n = 2$) and apply our attack method (see Algorithm 1) to 1,024 (for FMNIST and CIFAR) and 128 (for ImageNet) randomly selected test samples per pair. For each sample, the first backend uses the input's true class as its target, while the second backend is assigned a different, randomly chosen class.

The results of this experiment are displayed in Figure 5. We observe that Chimera examples can be consistently found across all backend combinations. The search on FMNIST yields relatively uniform success rates, ranging from $16\%$ to $24\%$. In contrast, both CIFAR and ImageNet exhibit more pronounced variation. We identify two clusters for both datasets: one consisting of all CPU-based backends, with success rates between $9\%$ and $18\%$ for CIFAR and $27\%$ to $44\%$ for ImageNet, and another induced by the cuBLAS GPU backend. When this backend is used, our method achieves significantly higher success rates of $95\%$–$96\%$ for CIFAR and $100\%$ for ImageNet.

We attribute these disparities to the convolutional layers in the CNN model used, which amplify numerical differences during GPU computation. Upon deeper inspection of the PyTorch code, we find that all CPU-based implementation appear to use the same default implementation for the convolution layer. For these backends, the differences only stem from the dense layers of the network and thus the error aggregation is significantly reduced. On the other hand, the CUDA-based GPU backend uses a separate convolution implementation. Due to the large amount of floating point operations in a convolution (Schlögl et al., 2024), the gap between the CPU backends and the GPU backend is significantly higher.

### 4.4. Ablation Study

An important component of our methodology is the quantization function $q$, which ensures that only vectors representing valid objects from $\mathbb{S}$ are returned as Chimera examples. In our search method, gradient computations are based exclusively on quantized inputs. To evaluate the effectiveness of this quantization step, we conduct an ablation study. Specifically, we compare our approach to an alternative strategy where we compute gradients on the unquantized inputs and only apply quantization to check whether a Chimera example has been found.

We find that the ablated setup perform worse. While our quantization step results in a slight performance reduction of about $1\%$ to $2\%$ for GPU-based backends, it yields substantial improvements for CPU-based backends, producing approximately $2.2\times$ more Chimera examples for CIFAR and between $1.1\times$ to $7\times$ more for FMNIST. As an additional benefit, the step saves one forward pass through the model, since only one pass on the quantized input is needed to obtain both classification results and gradients.

## 4.5. Alternative Approaches

We identify Chimera examples across all considered backend pairs, yet with varying success rates. This naturally raises the question: how well does our approach perform compared to alternative methods? To address this, we evaluate it against three baseline approaches.

*Binary Search.* Instead of following gradients, this baseline starts with two initial samples from different classes and iteratively interpolates between them until the resulting sample lies on the decision boundary.

*Boundary sample search.* Schlögl et al. (2021) propose a modified iterative FGSM that approaches the decision boundary. Their algorithm identifies "boundary samples", which are inputs used to fingerprint different CPU architectures and may therefore potentially constitute Chimera examples in our setting.

*Adversarial example search.* Carlini & Wagner (2017) suggest an attack that minimizes the perturbation applied to the original sample. Consequently, the resulting adversarial examples should be near the decision boundary and may serve as potential Chimera candidates.

To compare these baselines with our approach, we repeat the previous experiment and measure their respective success rates. Based on the observations from the previous experiment, we group the backends into two categories: CPU only backends and CPU vs. GPU backends. We report the mean success rates for both categories in Table 2.

We find that all three baseline approaches exhibit considerably lower success rates than our method across all datasets and categories. This difference is particularly pronounced for the FMNIST dataset, where the baseline methods either fail to find any Chimera examples or achieve success rates below 0.1 %. Similarly, for CPU-only backends, all baselines struggle to achieve good results on the CIFAR dataset, with the highest rate reaching 1.41 %. When considering the cuBLAS backend, baseline performance improves, with the method of Schlögl et al. (2021) achieving up to 67.38 %. Nonetheless, a significant gap remains between all baselines and our approach in every setting.

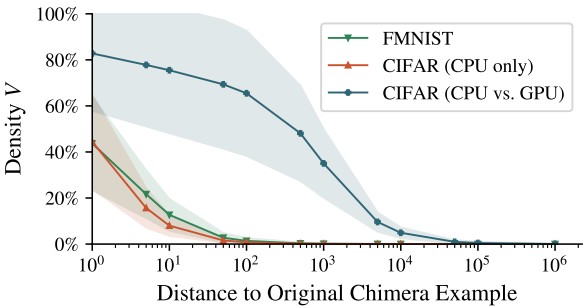

*Figure 6.* Estimated probability of infeasible Chimera examples around previously found Chimera examples at a given distance.

## 4.6. Spatial Analysis

So far, we know little about the characteristics of the regions containing the identified Chimera examples. Are these points isolated singularities that correspond to numerical anomalies between decision boundaries, or do they form "pockets" spanning hundreds of ULPs? While in both cases the phenomenon corresponds to only a tiny part of $\mathbb{F}$, in the latter, the vulnerability must be addressed on a broader scale to protect learning models from attacks.

To gain insight into this question, we broaden our setup: First, we lift the input constraints on the classification function $h$, allowing it to accept any vector from $\mathbb{F}$. Second, we relax our definition of Chimera examples to include any input from $\mathbb{F}$ that causes conflicting predictions. We refer to these as *infeasible* Chimera examples to distinguish them from those within $\mathbb{S}$. Our goal is to estimate the density of these conflicting vectors around a feasible Chimera example in the space of floats.

Due to the high dimensionality of $\mathbb{F}^d$, however, systematically traversing all adjacent vectors to some point is intractable. The curse of dimensionality makes exhaustive exploration computationally prohibitive. Consequently, we employ a sampling method to approximate the density of conflicting vectors at a distance of $d$ from a feasible Chimera example $\bar{x}$. Specifically, our sampling for a distance $d$ is defined as

$$V(\bar{x}, d) = \mathbb{E}_{\delta \sim \Delta_d} \left[ \mathbb{1} \left[ h_1(\bar{x} + \delta) \neq h_2(\bar{x} + \delta) \right] \right],$$

where $\Delta_d$ represents the probability distribution of perturbations with an $L_\infty$ norm of exactly $d$ ULP, and $\mathbb{1}(\cdot)$ is the indicator function.

To estimate $V$, we perform Monte Carlo sampling over the set of perturbations $\Delta_d$ and compute the average across all Chimera examples identified in previous experiments. By varying $d$, we can approximate the density of conflicting vectors surrounding a Chimera example, allowing us to assess the extent of the surrounding pocket, if one exists.

*Table 2.* Comparison of attack success rates for the baselines.

| | **FMNIST** | **CIFAR** | |
| --- | --- | --- | --- |
| | | CPU only | GPU/CPU |
| Ours | 19.19% | 13.61% | 95.26% |
| Binary search | < 0.1% | < 0.1% | 18.14% |
| Boundary samples | 0% | 1.41% | 67.38% |
| Adv. examples | 0% | 0% | 1.76% |

The results of this experiment are presented in Figure 6, where the x-axis denotes the distance $d$ to $\bar{x}$, and the y-axis indicates the density of conflicting vectors.

In this experiment, we rarely discover isolated Chimera examples. Instead, almost all examples are surrounded by additional conflicting vectors, forming pockets that range from 1 up to 100,000 ULP. While pocket sizes for CPU-based backends rarely exceed 100 ULP, the regions of conflicting vectors for GPU-based backends on CIFAR are considerably larger. This aligns with our experimental findings, where the success rate of our approach is significantly higher when one of the considered backends is Nvidia cuBLAS and the CIFAR dataset is used.

Although each of the considered pocket regions remains very small relative to the actual numerical values, the attack surface of Chimera examples spans a non-trivial region. Consequently, defenses must address this issue broadly, as we demonstrate in the following, rather than applying localized fixes to individual deviations.

## 5. Defense

Finally, we turn our focus to the challenge of preventing adversaries from discovering Chimera examples. Constructing defenses in adversarial machine learning is notoriously hard. Previous work has repeatedly shown that integrating robustness directly into models can be a tedious and often fruitless task (Athalye et al., 2018).

Since Chimera examples are closely related to adversarial examples, adversarial training (Goodfellow et al., 2015; Madry et al., 2018) appears to be a natural defense strategy. However, we find that it does not sufficiently reduce the attack success rate and can be further weakened by increasing the number of search iterations. Conceptually, this limitation arises because adversarial training is not designed to eliminate the tiny pockets of imprecision near the decision boundary, but instead shift the decision boundary. A detailed description of our setup and results is provided in Appendix A. As a remedy, we propose a defense based on a different principle: we introduce a secret to protect the model from Chimera examples without requiring any retraining or changes to the model architecture.

### 5.1. Keyed Noise Defense

Given a secret $s$, such as a random bit string known only to the defender, we introduce a *keyed noise function* $\zeta_d(x, s)$. This function generates uniform noise of magnitude $d$ ULP using a pseudorandom number generator (PRNG) initialized with the current input $x$ and the secret $s$. As a result, the generated noise remains deterministic for each pair $(x, s)$ but varies if either $x$ or $s$ changes.

Based on this primitive, we define a noisy version of the classification function as

$$h_i(x) = \arg\max_k f_i(x + \zeta_d(x, s))_k,$$

where each input $x$ is perturbed by noise of magnitude $d$ ULP, determined by $x$ and $s$. An adversary aware of the defense but lacking knowledge of $s$ cannot compute $\zeta_d(x, s)$ if a cryptographically secure PRNG is used. Since the noise depends on both $x$ and $s$, repeatedly running the same input through $h_i$ does not yield additional insights. At the same time, the prediction remains deterministic and does not interfere with existing machine learning workflows.

The parameter $d$ controls the strength of the added noise, thereby obstructing the discovery of Chimera examples in $\mathbb{F}$. Based on the results from the previous section, we find that these examples reside in confined regions. Thus, increasing the noise magnitude beyond the size of these regions significantly hinders their identification, as any attempt to gradually approach them will overshoot.

### 5.2. Defense Evaluation

To evaluate the defense, we assume an attacker with white-box access to the deployed target model, the utilized backends, and the noise magnitude of the defense; only the secret $s$ remains unknown. In Section 4.6, we observe that pockets for the FMNIST and CIFAR dataset extend at most to $10^3$ ULP and $10^5$ ULP, respectively. Based on these observations, we calibrate $d$ to these distances. Since this noise level is relatively small compared to natural input variations, we observe no negative impact on the test accuracy of the evaluated models.

To account for the additional noise layer, we extend the Chimera example search to include these noise perturbations. However, because the secret key $s$ is unknown, the adversary can only approximate this layer with their own key, with the goal to identify Chimera examples that remain resilient against the perturbations applied by the target model.

Using this adversary, we can evaluate the defense's effectiveness by repeating the experiment described in Section 4.3. The results are summarized in Table 3 and show a significant decline in success rate compared to the undefended model.

*Table 3.* Comparison of the attack success rate with and without the noise defense applied to the model.

| | FMNIST | CIFAR | |
| --- | --- | --- | --- |
| | | CPU only | GPU/CPU |
| w/o defense | 19.20 % | 18.04 % | 95.26 % |
| w/ defense | 0.00 % | 0.69 % | 0.00 % |

In most cases, the success rate drops to zero, with the only exception being CPU only backends on CIFAR, where we observe a minimal success rate of $0.69\%$. This rate could be further reduced by increasing the noise level $d$.

In summary, while our defense strategy does not eliminate the vulnerability, it introduces uncertainty, making it more difficult for an attacker to construct a successful attack. Even if an attacker were able to identify a Chimera example without directly approaching the decision boundary, the keyed noise function prevent them from exploiting it.

## 6. Related Work

Our work explores the combination of numerical imprecision in floats and adversarial machine learning. Consequently, it is closely related to previous research in these areas, which we briefly discuss in the following.

**Floating-point imprecision.** The existence of numerical deviations in floating-point arithmetic due to *swamping* is a long-standing issue (Goldberg, 1991). These errors have been experimentally studied in modern machine learning (Al-Rikabi & Renczes, 2022), paving the way for various attacks against neural networks.

For example, numerical errors in floating-point computations can compromise soundness assumptions in verifiably robust models (Jia & Rinard, 2021). They can also be exploited to embed neural backdoors during model pruning or through quantization artifacts (Tian et al., 2022). Similarly, subtle differences in model outputs can be leveraged to fingerprint CPU microarchitectures (Schlögl et al., 2021; 2024), as well as software stacks and GPU architectures (Zhang et al., 2024).

Our work extends this line of research by introducing linear algebra backends as another source of imprecision that when effectively exploited mislead models.

**Adversarial machine learning.** Our attack builds on existing work for generating adversarial examples. Technically, this research has primarily focused on two attack strategies. The first aims to minimize the required perturbation (Papernot et al., 2016; Carlini & Wagner, 2017) and improve efficacy across different models (Moosavi-Dezfooli et al., 2017; Gao et al., 2022). The second explores query-efficient algorithms under black-box threat models (Chen et al., 2020; 2017; Chen & Gu, 2020). Closest to our work is the method by Schlögl et al. (2021) that employs an iterative method to identify boundary samples for fingerprinting CPUs, which serves as a simple baseline in our study.

While our approach follows the blueprint of adversarial examples, it differentiates in two key aspects. First, we optimize over multiple slightly different functions whose internal workings are not differentiable, making successful attacks more challenging. Second, we construct feasible inputs adhering to the constraints of the input space $\mathbb{S}$, similar to problem-space attacks (Pierazzi et al., 2020).

## 7. Conclusions

Despite being largely invisible to practitioners, linear algebra backends are omnipresent in machine learning and introduce an unexpected vulnerability. In this work, we demonstrate that subtle implementation differences can be exploited to generate adversarial inputs, causing learning models to produce conflicting predictions depending on the backend used. These Chimera examples arise across all six backends examined in our experiments, revealing a previously unexplored attack surface in practice.

Searching for Chimera examples, however, is challenging due to the subtle and non-differentiable nature of underlying floating-point issues. The success rate of our attack varies, ranging from 8% for some backends to as high as 96% with Nvidia CUDA. Upon analyzing this phenomenon, we observe that numerical deviations between backends create "pockets" of 1–10000 ULPs near the decision boundary. Building on this insight, our defense prevents the reliable identification of pockets close to the boundary and thus impedes the search for Chimera examples.

## Impact Statement

We identify a novel attack surface in current machine learning systems and propose a corresponding defense. As a result, our work enhances the security of learning-based systems and mitigates potential risks to users.

## Acknowledgments

This work was supported by the Deutsche Forschungsgemeinschaft (DFG, German Research Foundation) under Germany's Excellence Strategy – EXC 2092 CASA – 390781972, and the project ALISON (492020528), as well as by the European Research Council (ERC) through the Consolidator Grant MALFOY (101043410).

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

## A. Adversarial Training as a Defense

Adversarial training (Goodfellow et al., 2015; Madry et al., 2018) is a defense that increases the robustness of models by incorporating knowledge about adversarial examples in the training process. Because of their close relation to Chimera examples, we perform an experiment to assess the effectiveness of adversarial training against our attack. Since training is performed on a single backend only, incorporating our attack into adversarial training is conceptually infeasible. As a remedy, we generate regular adversarial examples using projected gradient descent (Madry et al., 2018). We conduct this experiment on the FMNIST and CIFAR datasets.

We observe that the success rate of our attack is reduced by 30–40 % across the backends when adversarial training is employed. This is a moderate improvement but not as effective as our defense (see Section 5). We attribute this drop to our attack's initialization step, which leverages an adversarial example to reach the decision boundary and hence is obstructed by adversarial training. By increasing the number of iterations in this initial step, however, we can improve the attack's performance again. This highlights an interesting property of our attack: it initially behaves like a standard adversarial example to reach the decision boundary but then searches for Chimera examples in its vicinity.

## B. Dataset and Model Details

The evaluation of Chimera examples requires balancing resource-intensive model training with the need for experimentation across different hardware platforms, including less efficient CPU-based backends. Therefore, we select FashionMNIST (FMNIST) and CIFAR-10 (CIFAR) as our initial datasets for evaluation.

FMNIST (Xiao et al., 2017) is an improved alternative to the original MNIST benchmark and serves as our first evaluation dataset. It consists of 60,000 grayscale $28 \times 28$

images of fashion items for training and 10,000 for testing. For this dataset, we employ a neural network with two fully connected layers, each containing 128 neurons and separated by a ReLU activation function. The resulting model achieves an accuracy of 82.32% and comprises a total of 101,770 parameters.

CIFAR-10 (Krizhevsky et al., 2009) is a benchmark dataset consisting of color images of size $32 \times 32$ pixels, with 50,000 images for training and 10,000 for testing. We adopt a VGG-based model similar to variant $B$ in (Simonyan & Zisserman, 2015) for our experiments. Each convolutional block in our model consists of two $3 \times 3$ convolutional layers with padding and stride of one, followed by batch normalization, ReLU activation, and a 2D max-pooling layer. The model includes three such blocks, followed by three fully connected layers with 512 neurons each, ReLU activations, and dropout ($p = 0.5$). In total, the architecture contains 1,076,874 parameters and implements 9 of the original 13 parameterized layers from the VGG design—an appropriate scale for the relatively small dataset. After training for 10 epochs, the model achieves an accuracy of 80.75% on the test set.

As a more realistic benchmark, we additionally consider ImageNet (Deng et al., 2009), a large-scale dataset containing over 1.2 million images across 1,000 classes. For this setting, we conduct our experiments using a pre-trained EfficientNetV2-S model (Tan & Le, 2021), as summarized in Table A3. This architecture processes inputs through a series of bottleneck blocks, significantly reducing the number of trainable parameters. All inputs are resized to a consistent shape of $[b, 3, 224, 224]$ for inference. For clarity, Table A3 reports only the output shapes and parameter counts per block; we refer interested readers to the original paper (Tan & Le, 2021) for full architectural details. The model achieves a top-1 accuracy of 84.2% on the test set, placing it in the range of state-of-the-art models, such as CoCo, which reaches up to 91% accuracy (Yu et al., 2022).

*Table A1.* Our VGG-based architecture for CIFAR. Inputs are of shape ($b \times 32 \times 32 \times 3$) where $b$ is the batch size and the last dimension is the number of color channels.

| Block | Layer | Output Shape | Parameters |
|---|---|---|---|
| VGG Block 1 | Conv2D + BN + ReLU | [b, 128, 32, 32] | 3,840 |
| | Conv2D + BN + ReLU | [b, 128, 32, 32] | 147,840 |
| | MaxPool2D | [b, 128, 10, 10] | 0 |
| VGG Block 2 | Conv2D + BN + ReLU | [b, 128, 10, 10] | 147,840 |
| | Conv2D + BN + ReLU | [b, 128, 10, 10] | 147,840 |
| | MaxPool2D | [b, 128, 3, 3] | 0 |
| VGG Block 3 | Conv2D + BN + ReLU | [b, 128, 3, 3] | 147,840 |
| | Conv2D + BN + ReLU | [b, 128, 3, 3] | 147,840 |
| | MaxPool2D | [b, 128, 1, 1] | 0 |
| Flatten | | [b, 128] | 0 |
| Classifier | Linear + ReLU + Dropout | [b, 512] | 66,048 |
| | Linear + ReLU + Dropout | [b, 512] | 262,656 |
| | Linear | [b, 10] | 5,130 |
| **Total Parameters:** | | | 1,076,874 |

*Table A2.* Our DNN architecture for FMNIST. The ($b \times 28 \times 28 \times 1$) inputs are flattened and passed through a series of fully connected layers with batch size $b$.

| Block | Layer | Output Shape | Parameters |
|---|---|---|---|
| Flatten | | [b, 784] | 0 |
| Linear Block | Linear + ReLU | [b, 128] | 100,480 |
| | Linear | [b, 10] | 1,290 |
| **Total Parameters:** | | | 101,770 |

*Table A3.* EfficientNetV2S architecture for Imagenet.

| Block | Layer | Output Shape | Parameters |
|---|---|---|---|
| Stem | Conv2D + BN + SiLU | | 696 |
| | (Conv2D + BN + SiLU) x2 | [b, 24, 112, 112] | 10,464 |
| Block 1 | Conv2D + BN + SiLU + Conv2D + BN | | 25,632 |
| | (Conv2D + BN + SiLU + Conv2D + BN) x3 | | 124,416 |
| | | [b, 48, 56, 56] | 277,920 |
| Block 2 | Conv2D + BN + SiLU + Conv2D + BN | | 95,744 |
| | (Conv2D + BN + SiLU + Conv2D + BN) x3 | | |
| | | [b, 64, 28, 28] | 493,440 |
| MB-Blocks 1-30 | Conv2D + BN + SiLU | | 55,296 |
| | Conv2D + BN + SiLU | | 165,888 |
| | AvgPool2D + Conv2d + Conv2d + SiLU + Sigmoid | | |
| | Conv2d + BN | [b, 256, 7, 7] | 18,597,752 |
| Head | Conv2D + BN + SiLU | [b, 1280, 7, 7] | 330,240 |
| | AdaptiveAvgPool2D | [b, 1280, 1, 1] | 0 |
| Classifier | Dropout(0.2) | [b, 1280] | 0 |
| | Linear | [b, 1000] | 1,281,000 |
| **Total Parameters:** | | | 21,458,488 |

