# OpenReview forum: "Adversarial Inputs for Linear Algebra Backends"
_ICML.cc/2025/Conference — ICML 2025 poster_

### Official Review · Reviewer_QJ8E · 2025-03-04

**Overall Recommendation:** 4

**Summary:**

The authors propose a white-box attack to construct "Chimera examples", or inputs to models that elicit conflicting predictions depending on the employed backend library, and proposed a PRNG-based defense against it.

## update after rebuttal

The rebuttal addresses most of my concerns. In particular, I'm happy to see the authors running the experiment on ImageNet, demonstrating the scalability of their attack. I'm increasing my score.

**Claims And Evidence:**

1. It was claimed that Figure 1 contains Chimera examples, but the left figure has the same label "Truck" from BLIS and Apple Accelerate, which conflicts with the $\forall i \ne j$ requirement in Definition 1.
2. In section 4.4, it's unclear how many iterations are used by alternative attacks. Perhaps these attacks fail simply because you are using fewer iterations, not because they are inferior?

**Essential References Not Discussed:**

It's probably worth citing "Explaining and Harnessing Adversarial Examples" published in ICLR 2015, since it's the paper the proposed the fast gradient sign method (FGSM).

**Experimental Designs Or Analyses:**

The experiment uses very simple network architectures, i.e. three VGG block + three dense layer for CIFAR-10 and two fully
connected layers for FMNIST. Moreover, the networks have poor discriminative performance as "a test accuracy of 82.32% and 80.75%" is barely acceptable for datasets as simple as CIFAR-10 and FMNIST. It is unclear whether the attack also applies to more complex datasets and networks.

In the same vein, the defense proposed in section 5.2 has "no negative impact on the test accuracy of the evaluated models". Probably this is because the test accuracy is unimpressive to begin with. I imagine it's much harder to keep the test accuracy at 99%.

**Methods And Evaluation Criteria:**

Yes, it makes as much sense as prior works such as (Carlini & Wagner, 2017) and (Schlogl et al., 2021), on which this paper's method is based.

**Other Comments Or Suggestions:**

Typos:
1. Line 255, "liberary" should be "library"
2. Line 322 and 324, "\mathbb{F}" should be "\mathbb{F}^d"
3. Line 371, "up" should be "ULP"

**Other Strengths And Weaknesses:**

**Strengths**:
1. The attack surface is novel.
1. The ablation study in section 4.5 is helpful in understanding the landscape of Chimera examples.

**Weaknesses:**
1. The gradient-based attack algorithm does not seem novel. While there is a mapping $q: \mathbb{F}^d \to \mathbb{S}$, as I mentioned earlier there is no ablation experiments exploring its necessity.

**Questions For Authors:**

1. On line 168, what's the motivation for averaging across multiple backends?
2. On line 203, "The starting point $x_1$ is obtained from 2000 iterations of our search on a single backend to move towards the proximity of the decision boundary first;" why does this provide a good initialization?
3. On line 255, "To mitigate this effect, and to ensure the reproducibility of our experiments, we default to inference batches of size one." Why is the result deterministic when the batch size is 1? In that case, the backend/scheduler is still free to re-order computations due to the associative property of addition and multiplication.
4. On line 298, what's your explanation of the superiority of your method over alternative baselines? Your work is derived from the "Boundary sample search", so one may expect a similar performance.
5. In Figure 6, which dataset and network architecture are we talking about here?
6. In section 5.2, what if an attacker use a noisy gradient descent? Since considerable-sized clusters of Chimera examples exist, I imagine the attacker has a non-negligible chance of stumbling upon a Chimera example once it's close enough, i.e., when $x_k$ converges to the noisy ball.
7. It's surprising that Chimera examples exist, but it's unclear to me why they are harmful. Can you provide an example where they can be exploited to cause actual risk?

**Relation To Broader Scientific Literature:**

> So far, previous work has focused on floating-point imprecision arising from differences in CPU architectures, for
example, for fingerprinting systems (Schl ¨ogl et al., 2021; 2024) or breaking the certification of models (Jin et al.,
2022; 2024; Vor´aˇcek & Hein, 2023). Our analysis of linear algebra backends builds on this work; yet, we aim to induce significantly larger changes that flip the prediction of a model given an adversarial input. While differences in CPU architecture may further exacerbate this issue, we demonstrate that Chimera examples also exist between backends on the same CPU architecture.

This work identified a new attack surface, i.e., the difference in software implementations of linear algebra libraries on the same hardware. Moreover, I believe the different number of threads on the same BLAS backend can also affect the order in which floating point values are accumulated, and thus induce numerical imprecision, despite the fact that the authors didn't mention this in the paper. Maybe even different CUDA versions can induce some disparity, but I'm less confident about that.

**Theoretical Claims:**

It is claimed that
> If we dynamically adapt the step size $\alpha$, this approach theoretically brings us infinitesimally close to the decision
boundary, potentially leading to deviations among the backends.

This is not intuitive to me, so it would be great if the authors could provide a simple proof. I think the iteration should bring $x_n$ to a point with a large likelihood for $y_i$, instead of to the decision boundary.

Moreover,
> However, there is a catch: the generated points do not lie within $\mathbb{S}$, and thus moving along their gradients may lead into infeasible regions, as demonstrated in Section 4.4. To address this problem, we map $x_k$ back to $\mathbb{S}$ when computing its gradient, ensuring that the gradients reflect the view from $\mathbb{S}$ while optimization occurs in $\mathbb{F}$.

It is unclear to me how this address the problem, because we still have $x_{i+1} \not\in \mathbb{S}$, as illustrated in Figure 4. An ablation experiment without mapping $x_k$ back to $\mathbb{S}$ when computing its gradient would help clear some doubts.

---

> ### Author Rebuttal · Authors · 2025-03-31
>
> Thank you for your feedback on our paper!
>
> **Experiments with larger models.** We have extended our evaluation to include the ImageNet dataset, using the more complex architectures ResNet18 (Top-1 Accuracy: 69.7%) and EfficientNetV2S (Top-1 Accuracy: 84.2%). For these experiments, we used a reduced backend set (OpenBLAS, MKL, BLIS, and cuBLAS) and a smaller sample size of 128. In these new experiments, our attack achieves a 100% success rate for CPU-GPU backend combinations on both models. For CPU-CPU backends, we observe success rates of approximately 29% for ResNet18 and 22% for EfficientNetV2S. We will include these results in the paper.
>
> **Loss function in our attack.** Thank you for pointing out an issue in our presentation. Indeed, the description of the loss functions in Section 3.3 is imprecise, as it omits a key point: Following Definition 3.1, we define the target labels of the backends in opposition, such that always $y_i \neq y_j$. This represents an important distinction from the standard search for adversarial examples. Moving the input towards $y_i$ usually moves it away from $y_j$, and vice versa. This “tug of war” between the target labels effectively pulls the input closer to the decision boundary. We will add the precise definition of the target labels to our paper.
>
> **Role of quantization function.** We performed an ablation study to assess the influence of the quantization function on our attack. In this experiment, we compute the gradient of our attack without applying quantization. We find that our attack performs worse in this setting. Quantization is particularly important for CPU-based backends, yielding approximately 2.2× more Chimera samples for CIFAR and between 1.1× and 7× more for FMNIST. For GPU-based backends, we observe no such benefit. In fact, quantization slightly reduces performance by about 1-2%. We will include these results in our paper.
>
> **Intuition of search algorithm.** Our search algorithm brings us close to the decision boundary and then proceeds by alternating back and forth using the conflicting loss functions described above. Theoretically, for every gradient direction we take, there exists a point that lies exactly on the decision boundary and hence is potentially close to a Chimera example. By alternating through these directions in a tug-of-war manner, we converge toward the region where numerical deviations effectively determine the final decision.
>
> **Improvement over baselines.** All baselines were given the same iteration budget of 3000 steps as our attack, except for binary search that ran until convergence. Despite the same budget, our attack outperforms the baselines for the following reasons:
>
> 1. Although binary search can approach the decision boundary arbitrarily closely, it often lands in infeasible regions, making it impossible to elicit the required numerical differences from input samples.
> 2. The Carlini & Wagner attack was designed to target a single backend and searches for a point close to—but still across—the decision boundary. In contrast, our method explicitly targets the decision boundary and seeks inputs that yield conflicting predictions across backends.
> 3. The original boundary sample search, like binary search, was not designed to produce valid inputs. While it can find boundary points in feature space, the corresponding input samples often do not exist in the input domain.
>
> **Chimera examples in practice.** Chimera examples pose a threat whenever two different systems evaluate the same model on the same data. For instance, in forensic investigations, it is no longer sufficient to use only the same model and input to replicate an incident—such as a malware detection, the censoring of media content, or the decision of a hiring decision. Instead, the entire original system—including all backend libraries—must be fully replicated to ensure consistent and reliable results.
>
> **Noisy gradient descent.** When evaluating our defense, we already assume an attacker who is aware of the defense and uses noisy gradients. A key distinction of our defense is that the noise is deterministic and unique for each data point—that is, the noise remains fixed for the same input and key. As a result, an attacker cannot accumulate additional information by averaging over multiple runs. The noise creates a shattered view of the feature space, making fine-grained gradients ineffective except by chance. Our experiment in Section 5 shows that such chance-based success is too low to be practically viable.
>
> **Supplementary Material.** We have uploaded our source code to https://gitlab.com/anonymized-code/2025-icml and will make it available as open source to the community.
>
> **Image on first page.** You are absolutely right—this is a Chimera example with $n = 3$ only. We will replace it and plan to use an example from ImageNet to take advantage of the higher image resolution.

---

### Official Review · Reviewer_eV1D · 2025-03-11

**Overall Recommendation:** 4

**Summary:**

This paper investigates the vulnerability in neural network inference caused by minor discrepancies in linear algebra backends used by popular frameworks like TensorFlow and PyTorch. The authors introduce "Chimera examples," which are specially crafted inputs that produce conflicting predictions depending on the backend (e.g., Intel MKL, Nvidia CUDA, Apple Accelerate). These inputs exploit the inherent non-associativity of floating-point arithmetic and backend-specific optimizations that affect calculations subtly but significantly. The paper provides a comprehensive analysis of this vulnerability across several backends and proposes a defense mechanism to mitigate potential adversarial attacks exploiting these discrepancies. The findings highlight a novel attack surface within the machine learning pipeline that has been overlooked previously, emphasizing the need for robustness in backend implementations.

**Claims And Evidence:**

See strengths and weaknesses.

**Essential References Not Discussed:**

See strengths and weaknesses.

**Experimental Designs Or Analyses:**

See strengths and weaknesses.

**Methods And Evaluation Criteria:**

See strengths and weaknesses.

**Other Comments Or Suggestions:**

See strengths and weaknesses.

**Other Strengths And Weaknesses:**

Strengths:
1. Introducing the concept of Chimera examples that capitalize on backend-specific computational differences is a significant contribution to understanding security in machine learning systems. The paper extensively analyzes discrepancies across multiple major linear algebra backends, providing a broad view of the problem's scope.

2. Demonstrates the practical implications of theoretical discrepancies in backend computations, directly linking them to potential security vulnerabilities in deployed machine learning systems. Employs a rigorous methodology for generating and detecting Chimera examples, including detailed algorithmic strategies and adjustments for backend-specific characteristics.

3. Tests across a variety of platforms ensure that the findings are not limited to a specific hardware or software configuration, enhancing the generalizability of the results. Not only identifies a vulnerability but also proposes a novel defense mechanism, contributing both to the theoretical and practical aspects of machine learning security.

4. Addresses an immediate and practical concern in contemporary machine learning deployments, making the research highly relevant and timely. The experimental setup is robust, using popular datasets and architectures to validate the findings, which strengthens the paper's claims through empirical evidence. Provides detailed results including success rates of attacks across different setups, offering clear insights into the effectiveness of the proposed attack and defense strategies.

Weaknesses:
1. While innovative, the proposed defense mechanism is complex and may be challenging to implement in practice without affecting the system's efficiency or usability. The attack and defense strategies might be too tailored to the specific backends tested, which could limit their applicability in a broader range of environments or against future backend updates.

2. The threat model assumes white-box access to the model and backends, which might not always be practical in real-world scenarios, potentially limiting the applicability of the findings. The paper could benefit from a comparison with other types of adversarial attacks to position its contributions within the wider landscape of adversarial machine learning research.

3. The methods for detecting and defending against Chimera examples are likely resource-intensive, which could be a barrier for adoption in resource-constrained environments. It is unclear how the proposed methods scale with increasingly complex models or larger datasets, which is critical for modern deep learning applications.

4. The evaluation is somewhat limited to the datasets used (CIFAR-10 and FMNIST), and additional studies on different types of data might be necessary to fully understand the impacts. The paper does not fully explore how variations in experimental setups, such as different training regimes or model architectures, might affect the prevalence of Chimera examples.

**Questions For Authors:**

See strengths and weaknesses.

**Relation To Broader Scientific Literature:**

See strengths and weaknesses.

**Theoretical Claims:**

See strengths and weaknesses.

---

> ### Author Rebuttal · Authors · 2025-03-31
>
> Thank you for your feedback on our paper!
>
> **Experiments with larger models.** We have extended our evaluation to include the ImageNet dataset, using the more complex architectures ResNet18 (Top-1 Accuracy: 69.7%) and EfficientNetV2S (Top-1 Accuracy: 84.2%). For these experiments, we used a reduced backend set (OpenBLAS, MKL, BLIS, and cuBLAS) and a smaller sample size of 128. In these new experiments, our attack achieves a 100% success rate for CPU-GPU backend combinations on both models. For CPU-CPU backends, we observe success rates of approximately 29% for ResNet18 and 22% for EfficientNetV2S. We will include these results in the paper.
>
> **Efficiency and generality of our defense.** The computational complexity of the proposed defense depends only on the size of the input, not on the model itself. We leverage standard cryptographic libraries, which allow us to compute keyed noise for thousands of inputs per second. As a result, our defense adds less than 0.5% overhead to the overall inference process, making it practical for most application scenarios.
>
> In our defense, we deliberately avoid making any assumptions about the design or functionality of the employed backends. The only parameter we rely on is the empirically measured size of the pockets containing Chimera examples. A practitioner with knowledge of the specific backends used for a model can measure this size and configure our defense accordingly.
> However, you are correct that the required magnitude of defense noise depends on the complexity of the model, as more complex models typically accumulate larger discrepancies between backends during inference. Therefore, the noise level must be empirically calibrated on a per-model basis. Nonetheless, this does not affect the efficiency of our defense
>
> **White-box access.**  It is correct that our attack hinges on white-box access to the model and knowledge of the employed linear algebra backends, as shown in our experiments. However, finding Chimera examples is tricky, even in the white-box setting. Black-box attacks are, therefore, likely to suffer from low performance in practice. Keeping models confidential might, therefore, seem like a possible defense. We would still argue, however, that this is not a reliable protection strategy, as it requires keeping information confidential that is typically not considered secret. Moreover, in standard machine learning frameworks and environments, the available backends might be known to the adversary by default. We will acknowledge this setting in our paper.

---

> > ### Comment · Reviewer_eV1D · 2025-04-08
> >
> > Thanks for the author's rebuttal. The author's response did not solve all my questions well, so I kept my previous rating.

---

### Official Review · Reviewer_WNoZ · 2025-03-14

**Overall Recommendation:** 4

**Summary:**

The paper presents a method that exploits differences in the numerical computation implementations of linear algebra backends that power the major ML frameworks to construct adversarial examples.

**Claims And Evidence:**

Strengths:
- In the space of constructing adversarial examples, this paper is very novel and creative, which is a major strength of this paper.
- The result could potentially be used in areas of computer security that are normally not covered by adversarial examples, such as leaking which backend linear algebra implementation is being used just by querying predictions of the model.
- Method is simple and easy to execute.
- Results are sound.
- Paper is clearly written

Weaknesses:
- Simple defenses would likely prevent the attack, such as a slight randomization of the weights, or rejection of predictions that are too close to the boundary.
- Method is identical to the standard method for construction adversarial examples, but with a smaller step size, and thus the novelty of the  method is limited (but the application to attacking different linear algebra backends is novel).
- Limited evaluation, just two simple datasets for simple architectures.
- Appears to be most effective specifically when cuBLAS is used, and less effective for the other backends—what is special about cuBLAS?
- Transferability of the adversarial examples to other neural networks isn't evaluated, but would likely be 0.

Overall: I think the creativity of the submission plus the applicability to other areas of computer security I believe should lower the bar for its demonstrated applicability to real attack scenarios or the limited novelty of the actual method. For this reason, I am voting for acceptance.

**Essential References Not Discussed:**

See review.

**Experimental Designs Or Analyses:**

See review.

**Methods And Evaluation Criteria:**

See review.

**Other Comments Or Suggestions:**

See review.

**Other Strengths And Weaknesses:**

See review.

**Questions For Authors:**

See review.

**Relation To Broader Scientific Literature:**

See review.

**Theoretical Claims:**

See review.

---

> ### Author Rebuttal · Authors · 2025-03-31
>
> Thank you for your feedback on our paper!
>
> **Experiments with larger models.** We have extended our evaluation to include the ImageNet dataset, using the more complex architectures ResNet18 (Top-1 Accuracy: 69.7%) and EfficientNetV2S (Top-1 Accuracy: 84.2%). For these experiments, we used a reduced backend set (OpenBLAS, MKL, BLIS, and cuBLAS) and a smaller sample size of 128. In these new experiments, our attack achieves a 100% success rate for CPU-GPU backend combinations on both models. For CPU-CPU backends, we observe success rates of approximately 29% for ResNet18 and 22% for EfficientNetV2S. We will include these results in the paper.
>
> **Other defense strategies.** We initially considered different defense strategies but ultimately chose to randomize the inputs (rather than the weights) using keyed noise. This approach offers two advantages: First, the noise remains fixed for each input and, therefore, cannot be averaged out over multiple runs. Second, the runtime overhead of this defense does not depend on the model’s size and complexity. Note that we employ standard cryptographic libraries in our defense, which enables us to calculate thousands of inputs with keyed noise per second.
> An alternative strategy would be to reject predictions near the decision boundaries to prevent classifications. However, this comes with drawbacks: First, the margin around the decision boundary would still need to be chosen based on the pocket sizes of the backends. Second, introducing such a margin effectively creates a new attack surface for Chimera examples—this time between “rejected” and “accepted” predictions.
>
> **Analysis of cuBLAS.** We have further investigated the differences in attack performance between CPU-based and GPU-based backends. We found that the considered CPU-based backends employ identical implementations for the convolution operator. Hence, the numerical differences between them stem from matrix multiplication only. In contrast, the GPU-based backend cuBLAS uses a fundamentally different implementation for convolutions. As a result, numerical deviations arise from both convolution and matrix operations.
> As a result, we observe varying results for the FMNIST and CIFAR models. Since the FMNIST model consists solely of dense layers, the observed differences remain consistent across all backends. In contrast, the CIFAR (and ImageNet) models are largely composed of convolutional layers, which exhibit substantial variation when using cuBLAS. This variation makes it easier to identify Chimera examples in these models. We will include these results in our paper.

---

### Official Review · Reviewer_ava8 · 2025-03-20

**Overall Recommendation:** 1

**Summary:**

This paper claims that the implementations of linear algebra used by popular frameworks such as PyTorch and TensorFlow are not exactly consistent. The difference between these implementations can be quantified using a term called ULP (Unit in the Last Place). The authors demonstrate that this small gap is enough to produce adversarial examples specific to a given backend.

**Claims And Evidence:**

The contribution of this paper is somewhat unclear. Typically, the robustness of a model is measured by its accuracy against adversarial examples, which is often referred to as robust accuracy. However, this metric can fluctuate across different hardware, mathematical libraries, and objective functions. For example, AutoAttack and RobustBench use 11 different objective functions (one untargeted attack and ten targeted attacks) to generate adversarial examples. A model is considered robust only if it can defend against adversarial examples produced by all of these objective functions. This is a well-known observation in the field.

Given this, the authors should clarify (a) why the standard robust accuracy metric is not used in this paper, and (b) whether any specific backend exhibits a significant drop in robust accuracy compared to others. If no such drop is observed, the purpose of generating these adversarial examples remains unclear. I believe that the inconsistency of adversarial examples across various backends is not inherently problematic, as long as they adhere to standard definitions of adversarial examples. In other words, adversarial examples are not unique; multiple valid adversarial examples are an acceptable scenario.

**Essential References Not Discussed:**

[1] Towards Deep Learning Models Resistant to Adversarial

[2] Obfuscated Gradients Give a False Sense of Security

**Experimental Designs Or Analyses:**

* As mentioned earlier, the paper does not provide any experimental results on robust accuracy. Without these results, it is difficult to understand the significance of the adversarial examples presented by the authors.

* In Table 2, if I understand correctly, the attack success rates (ASR) refer to the ratio of "chimera" examples (not ordinary adversarial examples). It is unclear whether the term "Adv. example" in the last row refers to adversarial examples generated by the method described in the paper or by other means. Clarification is needed.

* Additionally, the target model used in the experiments is naturally trained, not adversarially trained. This suggests that generating adversarial examples should be relatively easy. In this context, the significance of these adversarial examples proposed by the authors remains uncertain.

* As shown in Table A1, Dropout layers are involved. Dropout is a stochastic process and should not be used during inference, as this would overestimate the robust accuracy.

* The proposed defensive algorithm appears to be ineffective. One can approximate the gradient through accumulation in the same way as described in the equation on line 167. For further details, please refer to the paper by [2].

**Methods And Evaluation Criteria:**

* As mentioned in the Summary section, robust accuracy is not reported in this paper. If this metric is not applicable, the authors should provide a convincing explanation to justify its exclusion to the reviewers and readers.

* I compared Algorithm 1 with the PGD attack proposed by Madry [1]. The first step projects the images onto a constrained set, and the second step calculates the efficient gradient by accumulating the gradients from each implementation. I would appreciate further clarification on whether there are any significant improvements in this approach compared to existing methods.

* The authors did not explain the reasoning behind the reparameterization of the input (line 180, right column). Could an adaptive attack achieve the same goal?

* The most important aspect of Algorithm 1 seems to be missing: the authors did not provide a clear definition of the projection function $\hat(x)_k = q(x_k)$. This should be clearly stated.

* The experimental configurations, including the radius of the epsilon ball, are not fully detailed. Providing more information on these settings would be beneficial.

**Other Comments Or Suggestions:**

*Please number all equations for clarity.

* In conclusion, I strongly suggest rejecting this paper due to unclear contributions and insufficient supporting evidence. The modifications required to meet the acceptance criteria seem substantial. However, I encourage the authors to provide more compelling evidence to support their claims. I would reconsider my recommendation if convincing revisions are made.

**Other Strengths And Weaknesses:**

A major weakness is the lack of adversarially trained models in the experiments.

**Questions For Authors:**

N/A

**Relation To Broader Scientific Literature:**

N/A

**Theoretical Claims:**

The paper lacks theoretical proofs. The concept of the infeasible area shown in Figure 3 appears overly simplistic. I would suggest that the authors provide a clear, formal mathematical definition of this area to strengthen the theoretical foundations of their work.

---

> ### Author Rebuttal · Authors · 2025-03-31
>
> Thank you for your feedback on our paper!
>
> **Chimera vs. adversarial examples.** We are sorry that a key distinction between Chimera examples and adversarial examples did not come across clearly. Constructing a Chimera example always requires considering at least two backends simultaneously. If only a single backend is attacked, the resulting adversarial example is likely to transfer to other backends as well—and thus does not qualify as a Chimera example. This is why standard algorithms for generating adversarial examples are typically unable to produce Chimera examples. As shown in Table 2, for instance, the attack by Carlini & Wagner (labeled Adv. Examples) failed to find a single Chimera example for the CPU-only backend pairs. We will clarify this point in the revised version of the paper.
>
> **Robust Accuracy.** We have considered different metrics for our evaluation but ultimately decided against using robust accuracy, as our focus is not on measuring changes in model robustness but on assessing the attack surface of backend pairs. Nonetheless, the presented attack success rate—shown in Figure 5 for different backend pairs—can be interpreted as a normalized version of (1 - robust accuracy). Since we introduce a new attack, we present this success rate, as it better highlights the efficacy of the attack rather than the resulting degradation in model performance.
>
> **Adversarial Training.** Thanks for this suggestion. Indeed, we had not considered adversarial training as a potential defense. However, this concept cannot be directly applied in our setting, as it would require training the same model across multiple backends simultaneously. Nonetheless, we conducted two experiments to investigate this defense in slightly different settings.
> First, we added Chimera examples to the training data of the considered models, which corresponds to a simplistic form of adversarial training. In this setting, however, we observe no impact on the success of our attack.
> Second, we performed adversarial training using a standard method for generating adversarial examples (PGD). Interestingly, this reduces the success rate of our attack by 30–40%. We attribute this drop to our attack’s initialization step, which leverages an adversarial example to reach the decision boundary. By increasing the number of iterations in this initial step, we can improve the attack’s performance again. This highlights an interesting property of our attack: it initially behaves like a standard adversarial example to reach the decision boundary but then searches for Chimera examples in its vicinity.
> We will include this additional experiment in the revised version of the paper. Still, our defense (Section 5) is capable of completely eliminating the attack without affecting model performance, unlike adversarial training.
>
> **Quantization and reparameterization.** The quantization function $q$ maps a vector $x$ in the feature space to the nearest element in a discrete set $S$ forming the input space. For instance, in the case of 8-bit images, it converts floating-point values in the feature space back to 8-bit pixel values. Similarly, reparameterization is a technique used to enforce box constraints on the input, ensuring that all pixel values remain within the valid range [0, 255]. This approach is derived from the Carlini-Wagner attack. We will clarify this in the paper.
> (Infeasible regions) Infeasible regions arise from the mismatch in granularity between the input space (e.g., 8-bit pixels) and the feature space (e.g., 32-bit floats). While Chimera examples may exist for certain combinations of 32-bit floating-point values, these may not necessarily be reachable using 8-bit inputs, hence lying in an infeasible region. We will clarify this difference and provide a more formal definition of this problem using the quantization function $q$.
>
> **Key differences.** Our attack differs from the classic PGD attack in two key aspects. First, we calculate the gradient from a quantized input in every iteration. Quantization is necessary because we want to elicit a Chimera example from a feasible (discrete) input. Second, the perturbation is computed from multiple linear algebra backends with conflicting loss functions. Each backend aims to push the input toward a different class—like in a tug-of-war scenario.
>
> **Defense effectiveness.** Our defense remains effective even when gradients are approximated or accumulated. It is effective because Chimera examples lie in extremely narrow regions of the feature space. As a result, adding noise with a magnitude larger than these regions during inference makes it very unlikely to discover them. As this noise is fixed for each input $x$ (keyed), it cannot be averaged out through repeated computations. Consequently, the computed gradients may move around the regions of Chimera examples but can only locate them by chance.
>
> **Drop-out layers.** We use drop-out layers only during training and not inference.

---

> > ### Comment · Reviewer_ava8 · 2025-04-02
> >
> > The responses provided are not convincing to me for the following reasons:
> >
> > * Unclear Motivation: First, from the perspective of an attacker, the goal is to minimize robust accuracy. However, robust accuracy is not reported in this paper. Second, it is well-known that robust accuracy can fluctuate across different hardware, mathematical libraries, and objective functions. The lack of study on this topic is simple: it is unnecessary from an attacker's standpoint. Third, the authors did not directly address my question regarding whether any specific backend exhibits a significant drop in robust accuracy compared to others. This is a crucial question. If ordinary adversarial attacks can significantly deceive models, I fail to understand why Chimera examples are worth studying.
> >
> > * Lack of Assessments on Adversarially Trained Models: If the authors wish to demonstrate that Chimera examples are important but have not been sufficiently explored, assessments on adversarially trained models should be included. As I mentioned in my previous comment, generating adversarial examples on standard trained models is relatively simple. The authors could easily download pre-trained adversarially trained models from RobustBench or other repositories to conduct experiments. If those models can defend against Chimera examples effectively, it would suggest that the proposed attack/defense is not meaningful. However, the authors decided to train very simple models as baselines, which is not a convincing assessment. I do not believe this evaluation accurately reflects real-world scenarios.
> > * Defense Method: The proposed defense, if I understand correctly, is aimed at black-box settings. However, numerous studies already exist on generating adversarial examples with imprecise gradient estimation, and the proposed defense does not seem to offer anything new in this regard.
> > * Limited Novelty: There are already many PGD-like attacks that incorporate specific constraints to generate custom adversarial examples. The key differences claimed by the authors do not present new concepts.
> > * Experimental Configuration: The authors' response does not fully detail the experimental configurations, including the radius of the epsilon ball, which raises concerns about the clarity and reproducibility of the experiments.
> >
> > I have carefully reviewed all the comments from other reviewers and the corresponding responses from the authors. However, based on the reasons outlined above, I maintain my original rating.

---

### Decision · Program_Chairs · 2025-05-01

**Decision:**

Accept (poster)

**Comment:**

This paper investigates a highly novel attack in neural network inference by exploiting subtle numerical discrepancies arising from different linear algebra backends (e.g., Intel MKL, Nvidia CUDA, Apple Accelerate). Specifically, this work introduces the concept of "Chimera examples": inputs crafted to produce conflicting predictions across backends. Additionally, a defense mechanism is provided to mitigate these attacks. The empirical evaluation across several datasets and models is provided to support its claim.

Overall, the reviewers strongly appreciate the strong novelty of this paper, as it brings attention to an underexplored vulnerability caused by backend-specific floating point differences. Additionally, reviewers are happy to see its practical implications under different experiment settings. But meanwhile, some concerns are raised, including: 1) larger models should be analyzed; 2) some other defense strategies (e.g., weight randomization) should be considered; 3) further analysis about cuBLAS is needed; 4) some key components of this attack algorithm (e.g., reparameterization, project function) need further clarifications; 5) its technical part is not significantly novel compared to prior PGD-like attacks.

The rebuttal addressed many of these concerns. Three reviewers ultimately voted positively, while Reviewer ava8 expressed reservations regarding low VGG accuracy, the absence of evaluations with adversarially trained models, and perceived limited novelty in the discussion. However, these points were rebutted by Reviewer WNoZ, who argued that the VGG variant used was smaller than standard and underscored the significant novelty and practical security implications of the study.

The AC concurs with Reviewer WNoZ and the other positive reviewers, recognizing the study as a valuable contribution that sheds new light on robustness from the perspective of linear algebra backend dependencies. Therefore, the AC recommends accepting this submission.